# Layout Design and Die Casting Using CAE Simulation for Household Appliances

**Hong-Kyu Kwon**

Department of Industrial and Management Engineering, Namseoul University, 91 DaeHakro, Seonghwan-eup, Cheonan-si 31020, Korea; hongkyuk@nsu.ac.kr; Tel.: +82-41-580-2206 or +82-10-3120-1390

**Abstract:** Due to the development and industrialization of science and technology, aluminum alloys have been developed in various fields. Recently, the government has been pursuing ways to decrease the weight and increase the recyclability of various components in order to conserve resources, energy, and the environmental. In keeping with this trend, cast iron products are being replaced by aluminum products in the foundry industry by using high-pressure die casting (HPDC). Casting layout design, relies on the experience and knowledge of mold designers in the casting industry, which proves insufficient to respond to the rapidly changing needs of the era and to increasing production costs. Designing and producing casting layouts using CAD/CAM/CAE technology has become a critical issue. Computer-Aided Engineering (CAE) technology is rapidly increasing with the development of computer software and hardware. CAE technology not only predicts defects in mass production but also performs filling or solidification analysis during the mold design stage before production, enabling optimal mold design methods. New technologies that combine the emerging casting processes of filling and solidification analysis using computer simulation with existing technology and practical experience in the field are rapidly increasing in the foundry industry. Based on empirical knowledge, the layout and design of casting products has traditionally progressed through trial and error. The solutions achieved through scientific calculation and analysis using CAE technology can save a great deal of money and time in the building of die-casting molds and in their design and fabrication. In this study, numerical analysis of household appliances (cooking grills) quickly and accurately predicts problems arising from the filling and solidification of the melted metal in the casting process, thereby ensuring the quality of the final cast product. These results can be used to quickly establish a sound casting layout with reduced production costs.

**Keywords:** gate system; high pressure die casting; CAE simulation; flow analysis; cooking grill

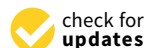



## 1. Introduction

Due to the development and industrialization of science and technology, aluminum alloys have developed in various fields. Due to recent problems involving resource shortage, energy conservation, and environmental problems, the manufacturing of cast iron and sand castings are being replaced by processes such as aluminum and high-pressure die casting (HPDC) [1,2].

Die-casting technology is an economical mass production technology that can manufacture complex parts at once and is a critical manufacturing technology in consumer goods manufacture. Its high dimensional stability and usefulness for mass production mean that it is gaining popularity as an optimal method for industries such as automotive parts and electronic components that require competitive high quality, low cost, and short delivery time. However, die-casting requires more advanced mold-making techniques due to the high temperature of the melt, high pressure on the mold surface, and the complexity and precision of the product shape [3,4].

CAE technology, widely applied in industrial sites, is widely used in the die-casting industry to address this problem. Generally, when manufacturing die-casting molds, they

are designed taking into consideration the layout of the molds and relationship with the extruders as well as the casting conditions, the design of the gating system, and the cooling conditions of the molds. Additionally, the degree and location of product defects caused by the casting process depends on the various casting methods. Recently, the advancement of CAE technology has dramatically reduced the existing trial-and-error process of creating casting molds. As a result, mold makers can produce high-quality casting parts, and production cost and time have been reduced [5–9].

Various casting process analyses have been constructed using CAE technology. A thin-walled notebook housing with less than 1mm thickness was investigated using computational solidification simulation [10]. The formability of bipolar plates for fuel cells was analyzed using computational filling behavior and solidification [11]; the castability of the optimized geometry, the absence of defects, and the local mechanical properties were verified and calculated using MAGMA software [12]. The various analyses for the optimal process design of an automobile gear housing was performed using filling behavior and solidification with MAGMA software [13]. The LPDC process parameters of a thin-walled component with less than 1.5 mm thickness were optimized using numerical simulation [14]. By using a computer-aided parametric design, the semi-automated design of a gating system was investigated to address the problem of the automatic generation of the gating elements and to utilize design evaluation [15].

In this study, CAE technology quickly and accurately predicts the filling and solidification processes during casting and establishes sound methods with reduced production costs. A cooking grill is a kitchen cooking utensil with a light weight and high thermal conductivity of ADC 12 types. Using CAE simulation software (AnyCasting, Seoul, Korea), three casting methods were reviewed to minimize the internal air porosities of castings and ensure quality stability. The aims were to analyze the mold filling and solidification processes in order to establish a defect control method and derive the optimal casting method for the die-casting mold design and fabrication process.

## 2. CAE Simulation of Die-Casting Process

Commercial packages (AnyCasting) developed by AnyCasting Co., Ltd. carried out the filling and solidification analysis using a hybrid numerical analysis method. This combined the PM (Porous Media) method and the Cut-Cell method to compensate for the shortcomings of the conventional FDM (finite difference method) rectangular mesh [16]. The CAE simulation analysis of the overall process, as can be seen in Figure 1, was separated by four groups: pre-processing, mesh-generation, simulation, and post-processing. Like other numerical analysis programs, AnyCasting converts feature models generated by 3D CAD commercial software into STL formats and uses them for pre-processing.

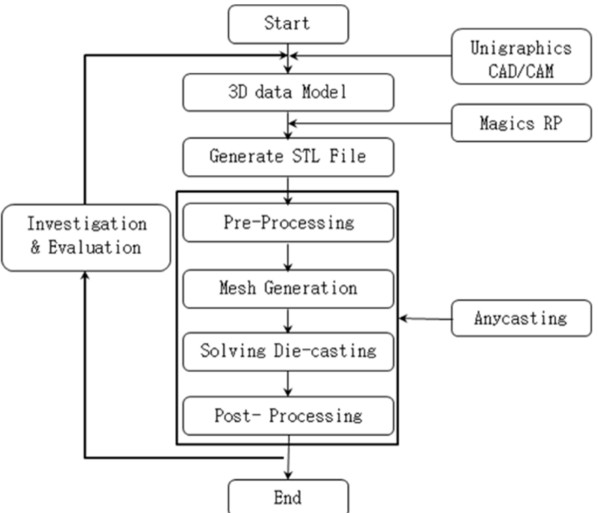

**Figure 1.** Flow Chart of CAE simulation.

### 2.1. Numerical Model of Die-Casting Process

In the die-casting process, the rapid motion of the chamber plunge generates rapid movement and flow of molten metal and charges the molten metal at high pressure in the mold cavity. AnyCasting is designed to analyze the flow of molten metal as a 3D fluid flow with free surfaces and boundaries using a hybrid method. There are three main phenomena, mold filling, solidification and cooling, and stress and strain distribution, to be considered in casting process modeling [17].

Fluid flow and heat transfer are described by mass balance, momentum balance, and energy conservation laws. According to the geometry of the melt flow path and the material properties, the following governing equations are non-linear. A set of simultaneous and algebraic equations are obtained by linearizing and discretizing them with numerical methods [7,9].

Continuity equation (when $T > T_S$):

$$\frac{\partial \rho}{\partial t} + \frac{\partial}{\partial x_j}(\rho U_j) = 0 \tag{1}$$

where $t$—time (s), $x$—space (m), $\rho$—density (Kg/m$^3$), $U$—velocity (m/s), $T$—temperature (C, K), and $T_s$—solid temperature (C, K).

Momentum equation (Navier–Stokes, when $T > T_S$):

$$\frac{\partial}{\partial t}(\rho U_i) + \frac{\partial}{\partial x_j}(\rho U_j U_i) = \frac{\partial \rho}{\partial x_i} + \frac{\partial}{\partial x_j}\left(\mu \frac{\partial U_i}{\partial x_j}\right) + \rho g_i \tag{2}$$

where $t$—time (s), $x$—space (m), $\rho$—density (Kg/m$^3$), $u$—kinematic viscosity (m$^2$/s), $g$—gravity (Kgf), $U$—velocity (m/s), $T$—temperature (C, K), and $T_s$—solid temperature (C, K).

Volume of Fluid (VOF) for the open surface:

$$\frac{\partial F}{\partial t} + U_j \frac{\partial F}{\partial x_j} = 0,\ 0 \le F \le 1 \tag{3}$$

where $t$—time (s), $x$—space (m), $F$—volume (m$^3$), and $U$—velocity (m/s).

The Fourier heat conduct equation is used for solidification modeling. Since heat flows from high temperature to low temperature, the movement of heat via heat conduction follows Fourier's law [7]. Solidification is a complex problem of phase transformation involving the absorption and release of heat. This feature has nonlinear boundary conditions between different phases, making it impossible to interpret the temperature distribution without including many assumptions (uniform physical properties, contact thermal resistance of phase change, thermal resistance of mold boundary) accurately [18]. The following conservation equations can describe the solidification modeling of molten metal on the condition that the heat balance and distribution of castings are laminar fluid flow and heat transfer processes [19,20].

Energy equation:

$$\frac{\partial}{\partial t}(\rho C_p T) + \frac{\partial}{\partial x_j}(\rho C_p U_j T) = \frac{\partial}{\partial x_j}\left(\lambda \frac{\partial T}{\partial x_j}\right) + Q \tag{4}$$

where $t$—time (s), $x$—space (m), $\rho$—density (Kg/m$^3$), $C_p$—heat capacity (J/K), $\lambda$—conductivity (W/m$^2$K), $U$—velocity (m/s), $T$—temperature (C, K), and $Q$—heat source (C, K).

Governing Differential Equation (FDE):

$$\frac{\partial(\rho \varphi)}{\partial t} + \frac{\partial(\rho U_j \varphi)}{\partial x_j} = \frac{\partial}{\partial x_j}\left(\Gamma_\varphi \frac{\partial \varphi}{\partial x_j}\right) + S_\varphi \tag{5}$$

where $\varphi$—generated variables which denotes mass, momentum, volume fraction of fluid and energy conservation, *t*—time (s), *x*—space (m), $\rho$—density (Kg/m$^3$), *U*—velocity (m/s), $\Gamma_\varphi$—generated diffusion coefficient, and $S_\varphi$—source term.

### 2.2. Geometry Model of CAE Simulation

Using 3D CAD software (Unigraphics NX10), the 3D solid modeling of cooking grills applied in this study was designed and then converted to STL files using 3D CAD software (Magics RP), as in Figure 1. During pre-processing, the converted STL files formed a group of materials such as castings, overflows, grating, gating, and molds.

As shown in Figure 2, the casting layouts applied in this study were set up in three ways. Casting layout 1 installed eight in-gates and two group overflows, while casting layout 2 installed one ring gate and two group overflows. Casting layout 3 was designed with the same shape as casting layout 2 on the fixed side, with a different profile than casting layout 2 on the moving side.

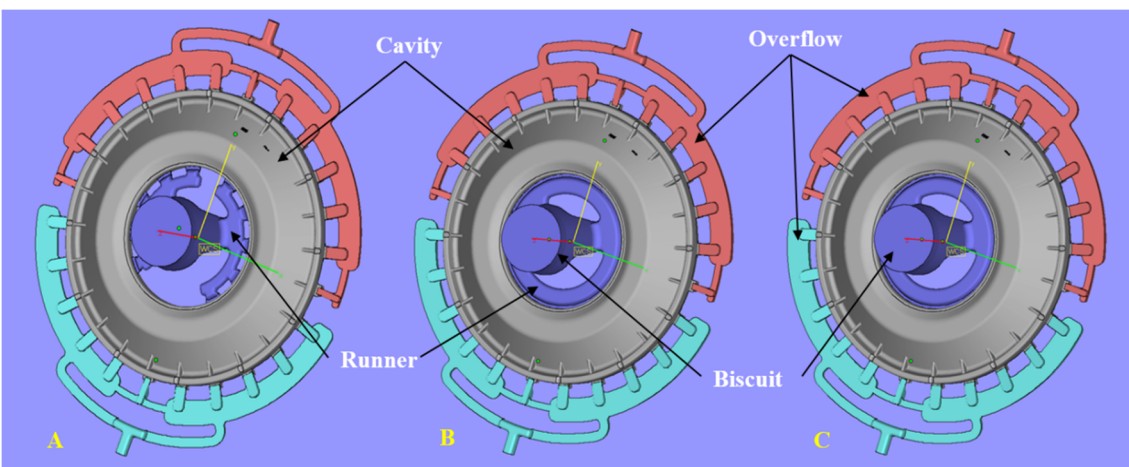

**Figure 2.** Casting model of cooking grill: (**A**) Case 1; (**B**) Case 2; (**C**) Case 3.

### 2.3. Condition of CAE Simulation

The interpretation conditions applied to the casting analysis are shown in Table 1; the casting material applied to the casting analysis is ADC12 (AlSi9Cu3) in Table 2. The mold material was SKD61 type. The initial warm-up temperature was set as 2000c, the mold during casting as 2800c, and the initial temperature of the melt material as 6400c. The die-casting machine was a cold chamber type with a clamping force of 850 tons; the plunger was 70 mm in diameter, 0.2 m/s at low injection speed, and 6.0 m/s at high injection speed. In the mesh generation process, mesh generation was achieved by varied interval element segmentation; the total number of meshes used for analysis was 12,497,000.

**Table 1.** Condition for the CAE simulation.

| Part | | Mold | | Plunger | |
|---|---|---|---|---|---|
| Material | ACD12 | Material | SKD61 | Diameter | 70 mm |
| Liquidus Line | 580 °C | Initial Temperature | 200 °C | Slow Velocity | 0.20 m/s |
| Solidus Line | 515 °C | Casting Temperature | 280 °C | High Velocity | 6.00 m/s |
| Initial Temperature | 640 °C | | | Length | 510 mm |
| Weight for casting | 1087 g | | | | |

**Table 2.** Chemical composition (%) of ADC12.

| C | Si | Mn | P | S | Cr | Ni | Mo | V |
|---|---|---|---|---|---|---|---|---|
| 0.32–0.42 | 0.80–1.20 | <0.50 | <0.03 | <0.03 | 4.50–5.60 | | 1.00–1.50 | 0.80–1.20 |
| **W** | **N** | **Cu** | **Co** | **Pb** | **B** | **Nb** | **Al** | **other** |

## 3. Results and Discussion

A gating system consists of runners, biscuits (sprues), and a gating system. The gating system is a casting layout design that can smoothly fill the melt material into the mold cavity to obtain the complete product. In addition, the gating system's design is one of the most influential factors in the overall problem of casting and the quality of the casting. Therefore, if the gating system design is inadequate and the defect rate is high, significant modification of the mold and in some cases its total replacement may be a required [13,15,21].

The flow velocity of the molten metal is a significant factor affecting the quality of the product and the life of the mold in the filling process. If the filling speed is too slow, it causes defects such as misrun and cold flow due to high heat loss, while if the filling speed is too fast, it reduces mold life by promoting wear on the runners and in-gates of the mold cavity [6,9,13].

### 3.1. Flow Analysis of CAE Simulation

Figure 3 represents the melt flow during filling by the flow analysis results for each casting layout. As mentioned in the previous section's interpretation conditions, the two-stage injection rate was established empirically considering the optimal mass production conditions and characteristics of similar products. To minimize the product's internal porosity, the speed switching point from low speed (0.20 m/s) to high speed (6.00 m/s) was set when about 5% of the mold cavity progress was completed after filling the runner and ingates, as shown in Figure 3.

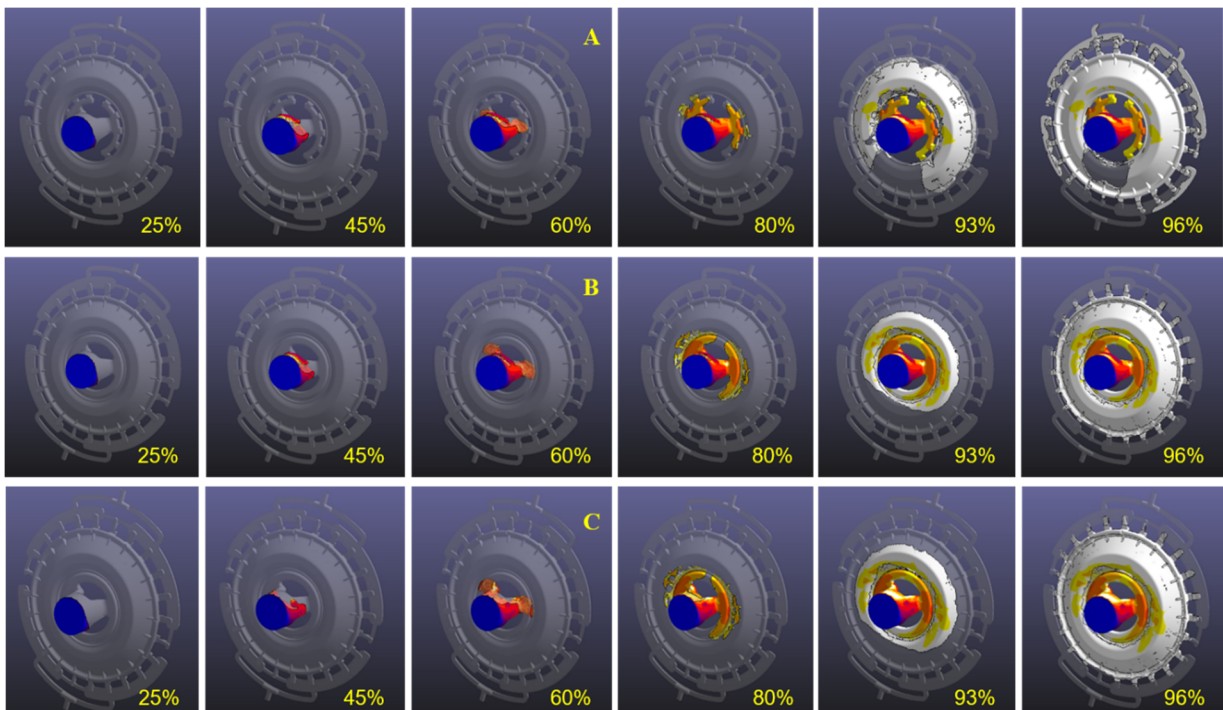

**Figure 3.** Simulation results of the melt filling process: (**A**) Case 1; (**B**) Case 2; (**C**) Case 3.

As shown in Figure 3, the filling is smooth without misruns or cold shuts of the molten metal. In cases 2 and 3, the melting metal flow can be observed uniformly compared to casting layout 1. It can be seen that the location of the isolating internal porosity that can occur when filling is less in casting layouts 2 and 3 than in casting layout 1.

### 3.1.1. Flow Analysis of CASE 1

Figure 4 shows the results of 93% and 96% filling behavior in casting layout 1. The filling behavior of casting layout 1 is imbalanced, as shown in Figure 4A. As shown in the 96% filling behavior of Figure 4B, it can be expected that the filling behavior with the imbalance results in air porosity isolation. Therefore, it was deemed that casting layout 1 was not appropriate. Based on the results of casting layout 1, casting layout 2 of Figure 3B was designed.

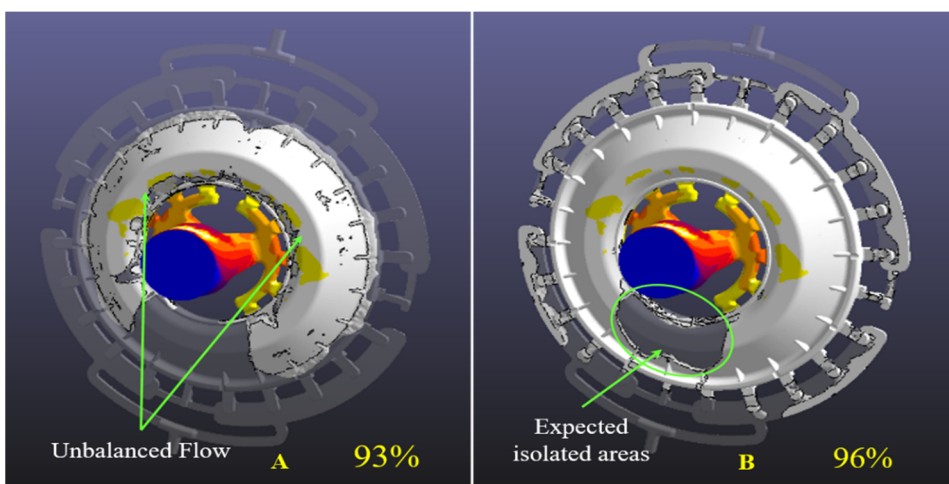

**Figure 4.** Simulation results of the melt flow for Case 1: (**A**) 93% filling; (**B**) 96% filling.

Figure 5 shows 96% filling behavior in casting layouts 1 and 2. Due to the design change of the gating system, the melting flow is improved, and the filling behavior is balanced. Hence, the isolated area of the air porosity disappears in casting layout 2. Therefore, it was judged that casting layout 2 is superior to casting layout 1.

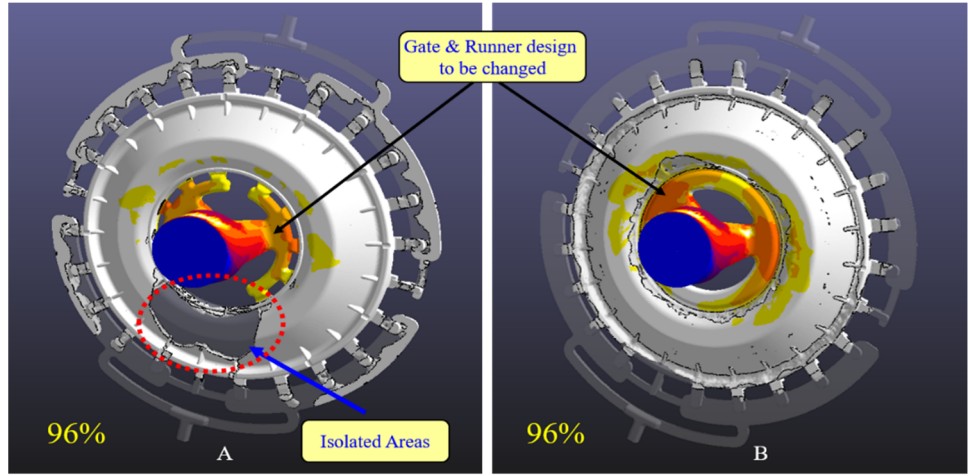

**Figure 5.** Simulation results of the melt flow: (**A**) 96% filling of case 1; (**B**) 96% filling of case 2.

### 3.1.2. Flow Analysis of Case 2

Figure 6 shows the results for 45% and 60% filling behavior of casting layout 2. Vortex shapes in biscuits (sprues) appear in filling behavior at 45% and 60%. In the early

stages of filling behavior, the backflow phenomenon is expected to result in a current phenomenon. Based on the results of Figure 6, it was also deemed that casting layout 2 was not appropriate, and improvement of the casting layout was derived. Based on the above results, casting layout 3 in Figure 3C was designed. Figure 7 shows the design features for the operating surfaces of casting layouts 2 and 3. As shown in Figure 7B, a round feature was added to the biscuit's lower center. The round shape was expected to reduce the reflux of the flow at the beginning of filling and reduce the eddy phenomenon that can occur during the filling process due to elution flow.

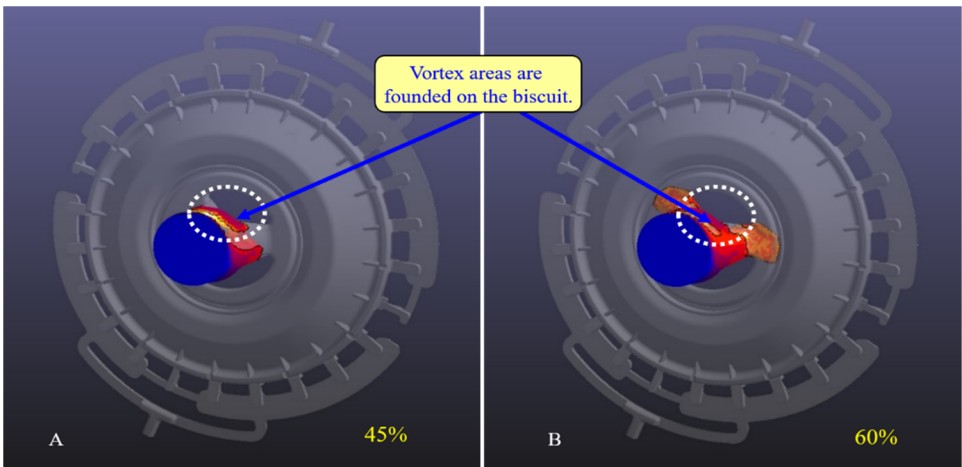

**Figure 6.** Simulation results of the melt flow for case 2: (**A**) 45% filling; (**B**) 60% filling.

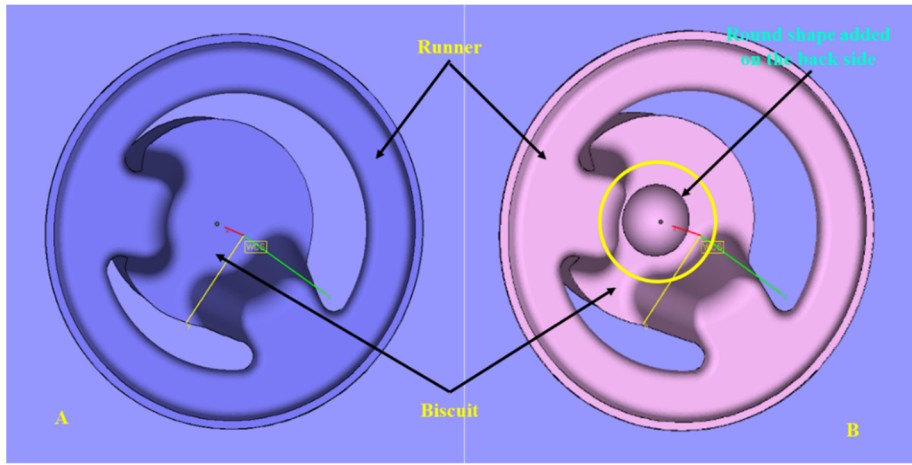

**Figure 7.** Gate system: (**A**) case 2; (**B**) case 3.

Figure 8 shows the results for 45% and 60% filling behavior of casting layout 3. It was deemed that the design change of the casting layout reduced the backflow in the early filling behavior, and reduced the eddy phenomenon. Based on the results on filling behavior in Figure 8, it was deemed that casting layout 3 was the most appropriate casting plan.

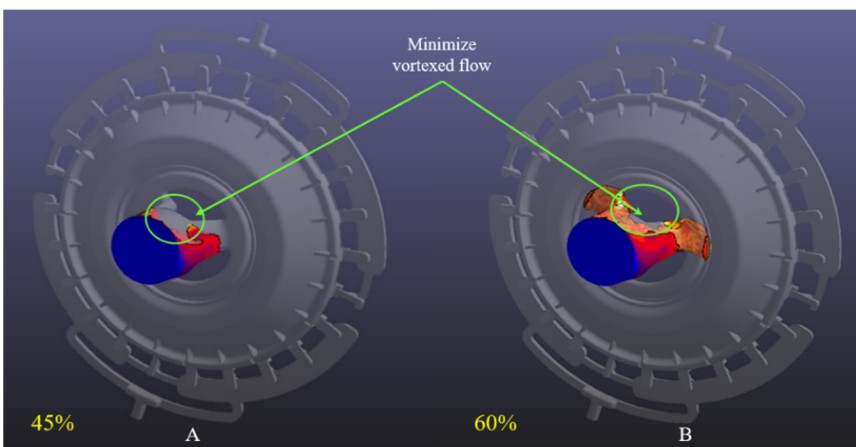

**Figure 8.** Simulation results of the melt flow for Case 3: (**A**) 45% filling; (**B**) 60% filling.

*3.2. Solidification Analysis of CAE Simulation*

Solidification is a complex problem of phase change involving the absorption and release of heat. It has different temperature gradients near the temperature range involving the phase change, making it difficult to access the mathematical interpretation. Because of this problem, various approximate solutions and numerical analyses attempt to interpret the solidification phenomenon [18].

Solidification analysis aims to estimate the location and degree of shrinkage porosity defects, for which various prediction methods are used. Shrinkage porosities are caused by the decrease in volume when most castings cause a phase change. In other words, when the casting changes from liquid to solid, the volume becomes less when it is solid than it was as a liquid. This volume reduction rate is around 4% to 6% in zinc, aluminum, and magnesium. This area, or hot spot, appears in many parts and is called shrinkage porosity when generated inside the casting [22].

The solidification analysis predicts where shrinkage defects are most likely to occur and predicts the cooling temperature of the casting product. The shrinkage defects mainly occurred on the thick area of the casting product. As shown in Figure 9, solidification analysis takes place after a filling is completed. The effect of cooling conditions is not very visible at the early stage. The expected solidification shrinkage areas are shown in 65% of the analysis results of casting layout 3.

Figure 10 represents several sections of the expected shrinkage defects. In the solidification analysis, the modular method of AnyCasting observed the shrinkage defect areas in several sections. One way to deal with shrinkage defects is to change the position of the local hot spot, which can be done by controlling the mold temperature. In thin casts, the process of changing the mold temperature is more manageable for changing the hot spot position, while in thick casts, it is much more difficult.

Another critical factor in preventing shrinkage defects is to keep the temperature inside the casting as uniform as possible. The mold temperature must be uneven to maintain this condition inside the casting. For example, the mold should be cold in the thick cast area, while in the thin cast area it should be hot. As a result, the casting temperature becomes much more uniform during solidification, resulting in the dispersion of shrinkage porosities. Therefore, a detailed cooling design should be added to the solidification shrinkage defect area during the mold design and fabrication process.

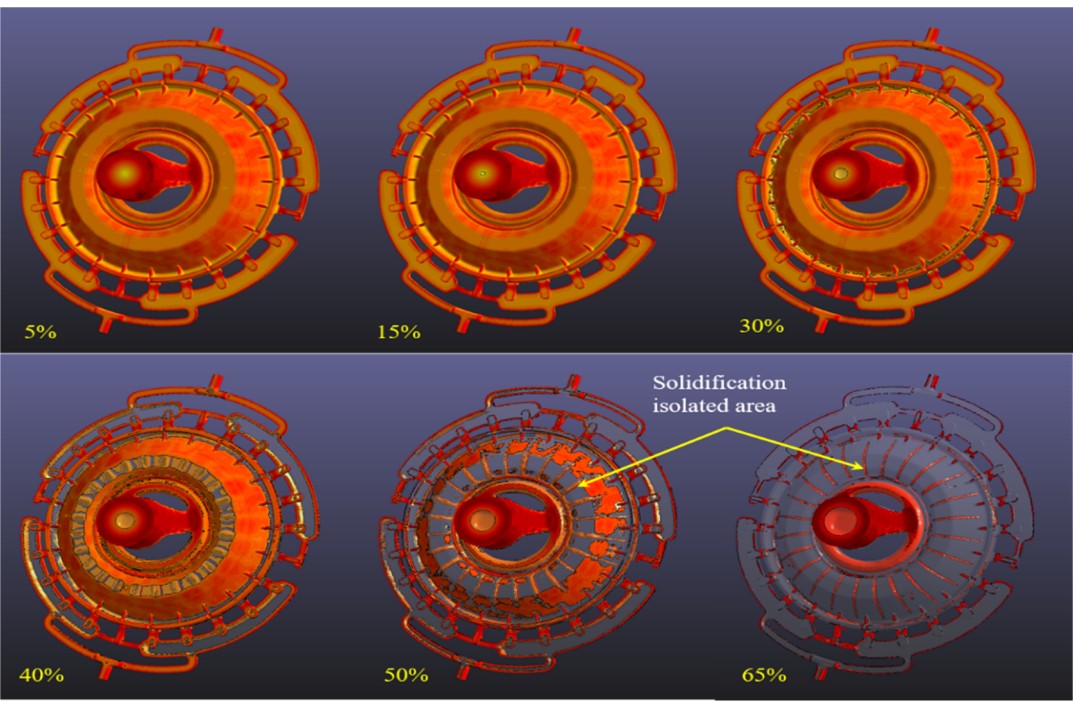

**Figure 9.** Simulation results of the solidification for case 3.

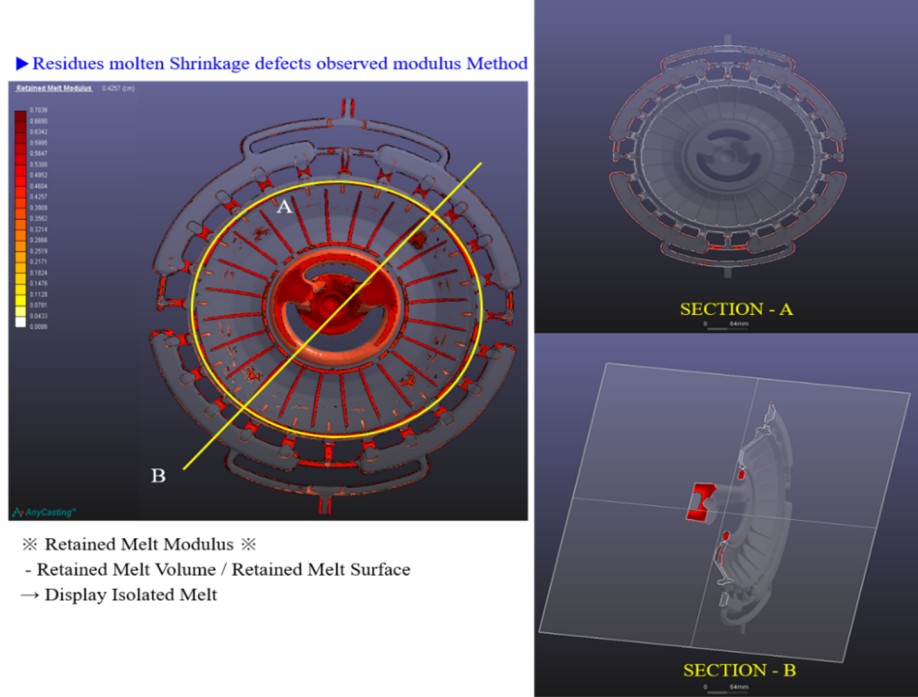

**Figure 10.** Expected shrinkage defect area for Case 3 at 65% solidification.

## 4. Improvements of the Casting Layout

Flow analysis and solidification analysis represents the best results of casting layout 3 compared to the other casting layouts. In the flow analysis results of the final casting layout shown in Figure 11, the presence of fine porosity has been observed on the moving side of casting layout 3. It is believed that the problem of microporosity will be significantly improved by improving the casting conditions during casting operations, allowing the

flow of the molten metal to be more uniform and cooling to be more meticulous during mold production.

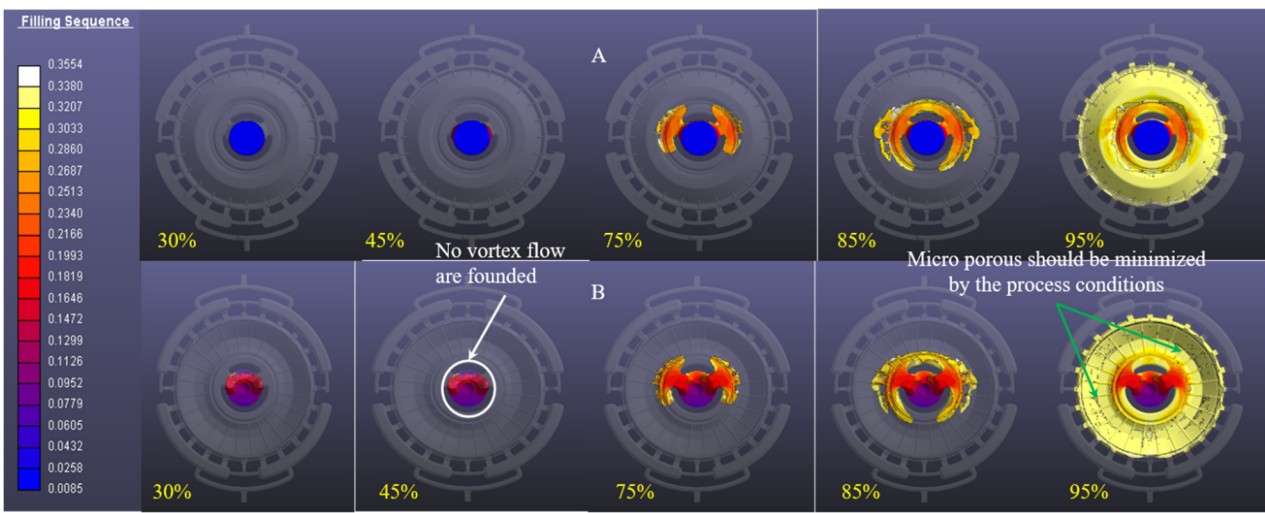

**Figure 11.** Simulation results with the final casting layout: (**A**) Fixed side; (**B**) Moving side.

## 5. Conclusions

In general, two critical factors validate the effectiveness of our simulation results: first, the experience of the previous project can be applied to subsequent projects, and second, the procedures of modeling and simulation are based on general physical rules [7,19,23]. In high-pressure die casting, modeling is the production of casting procedures in simulation programs. Using this method, almost all detailed process flows are determined as boundary conditions for computation. As a result, mold filling, solidification, microstructure and formation of properties, and the deformation of casting parts can be viewed. Die casting modeling and simulation become more critical because they are the fastest and most cost-effective ways to develop high-pressure die casting products [2,13,17,23].

Flow analysis with various casting layouts of, in this case, a cooking grill can be achieved by using the AnyCasting flow analysis program:

(1) According to the flow analysis results, the location of air isolation and the direction of filling of the molten flow was adequately determined. In addition, the shape of the most appropriate in-gate was appropriately identified according to the several casting layouts. It was determined that the results can be used to properly eliminate various problems caused by an imbalance in the melt flow.

(2) According to the flow analysis results, the melt flow in the final casting, layout 3, was the most uniform than compared to casting layouts 1 and 2. Solidification analysis of the final casting layout also identified microshrinkage during the solidification process. It is believed that securing adequate cooling locations in the mold development process will significantly reduce trial and error of different types and shorten the mold development period.

(3) According to the flow analysis results, the reflux phenomenon caused by backflow during filling was initially predicted, and a method was found to equalize the melt flow. Applying these results can reduce the reflux phenomenon when filling, and save time and expense of casting post-processing.

**Funding:** Funding for this paper was provided by Namseoul University.

**Institutional Review Board Statement:** Not applicable.

**Informed Consent Statement:** Not applicable.

**Conflicts of Interest:** The author declares no conflict of interest.

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
