# Peer review of "Layout Design and Die Casting Using CAE Simulation for Household Appliances"

_applsci, doi:10.3390/app112110128_

Round 1
Reviewer 1 Report
In this MS, the author used a commercial software to model die casting of three variants representing a common household appliance component and results from the model runs were presented with qualitative observations made from the visual results.
There are minor grammatical issues, which can be corrected with rigorous proofreading.
However, there scientific or engineering contents is lacking on the MS. The cursory introduction of the governing equations neglected additional, more complex processes such as phase change (solidification) and subsequent flow in the partially solidified porous structure.
The methodology presented on the MS is consistent with the typical design-simulation-analysis workflow for CAE. Without comparison with observation, it is very difficult to judge the skill of the model. Also, the presentation of the results consists of a series of 3D plots. While there are some interesting observations, for instance, the development of (allegedly) mono-porous zones in one of the designs, the remaining results are a simple regurgitation of the visual simulation results with little analytical work.
This MS is of very low scientific or engineering merit, and is one the reviewer would expect to see in CAE software user conferences. It is therefore recommendation of the reviewer that it be rejected.
Author Response
In this MS, the author used a commercial software to model die casting of three variants representing a common household appliance component and results from the model runs were presented with qualitative observations made from the visual results.
Point 1: There are minor grammatical issues, which can be corrected with rigorous proofreading.
Point 1 Response: The paper is reviewed carefully to improve the English quality. On your point, English language editing is still needed. I will request the English language editing from MDPI to improve the English qualification of the article after clearing other issues.
Point 2: However, there scientific or engineering contents is lacking on the MS. The cursory introduction of the governing equations neglected additional, more complex processes such as phase change (solidification) and subsequent flow in the partially solidified porous structure.
Point 2 Response: Thank you for your comment. Yes, some modifications are added in the introduction and sections 2.1 and 3.2 to improve the paper quality.
Point 3: The methodology presented on the MS is consistent with the typical design-simulation-analysis workflow for CAE. Without comparison with observation, it is very difficult to judge the skill of the model. Also, the presentation of the results consists of a series of 3D plots. While there are some interesting observations, for instance, the development of (allegedly) mono-porous zones in one of the designs, the remaining results are a simple regurgitation of the visual simulation results with little analytical work.
Point 3 Response: Thank you for your comment. Yes, it seems to be a typical CAE process and has some limitations and weak points of the paper. It would be better to convince our results if I provided the actual casting parts, X-ray, or CT measurements. However, it is too much cost for us to fabricate a mold to validate the simulation results. From my previous experience of other projects, the simulation result itself seems to be enough information for a mold-maker to save cost and time while producing the casting mold.
Point 4: This MS is of very low scientific or engineering merit, and is one the reviewer would expect to see in CAE software user conferences. It is therefore recommendation of the reviewer that it be rejected.
Point 4 Response: Thank you for your comment. To improve the paper's quality, some modifications are added in the introduction, sections 2.1 and 3.2, and the conclusion. I am well aware of the shortcomings and weaknesses of the paper. Please review the revised manuscript generously again.

Reviewer 2 Report
In this paper, the authors have used CAE simulation to optimise the design of casting layout. The study is very interesting. However, several concerns in the current stage of manuscript need to be addressed before consideration of acceptance, as follows:
To make the manuscript self-contained and more readable, make sure all the variables have been fully specified after each equation.
Please add more discussion and limitations of the proposed method.
More discussions and literatures should be added in the introduction.
Is there any other limitation of the proposed idea? Such as the specific or limited ranges of working environment or conditions? Is the proposed architecture still valid under other environment?
Carefully recheck grammar and typo errors.
Author Response
In this paper, the authors have used CAE simulation to optimize the design of casting layout. The study is very interesting. However, several concerns in the current stage of manuscript need to be addressed before consideration of acceptance, as follows:
Point 1: To make the manuscript self-contained and more readable, make sure all the variables have been fully specified after each equation.
Point 1 Response: Thank you for your comment. Yes, there are some modifications added in section 2.1, as you did request.
Point 2: Please add more discussion and limitations of the proposed method.
Point 2 Response: Thank you for your comment. To improve the paper's quality, some modifications are added in the introduction, section 3.2, and the conclusion.
Point 3: More discussions and literatures should be added in the introduction.
Point 3 Response: Thank you for your comment. With some modifications added in the introduction, section 3.2, and the conclusion, some references are enough added.
Point 4: Is there any other limitation of the proposed idea? Such as the specific or limited ranges of working environment or conditions? Is the proposed architecture still valid under other environment?
Point 4 Response: Thank you for your comment. Yes, there are some limitations and weak points of the paper. It would be better to convince our results if I provided the actual casting parts, X-ray, or CT measurements. However, it is too much cost for us to fabricate a mold to validate the simulation results. From my previous experience of other projects, the simulation result itself seems to be enough information for a mold-maker to save cost and time while producing the casting mold.
Point 5: Carefully recheck grammar and typo errors.
Point 5 Response: The paper is reviewed carefully to improve the English quality. On your point, English language editing is still needed. I will request the English language editing from MDPI to improve the English qualification of the article after clearing other issues.
Reviewer 3 Report
The paper presents a CAD/CAM/CAE analysis of high-pressure die casting process using a kitchen cooking grill as case study. Standard 3D techniques (UNIGRAPHICS and MAGIC) are applied for generating three variants of runners which are further analyzed by commercially available AnyCasting software regarding fluid flow, filling and solidification process. The comparison of three variants is correct and the advantages of third variant against others is clearly documented. The value of the paper is the good application of available tools for a given task, but without any scientific added value and experimental validation. I am suggesting for the authors to extend the paper with experimental results of optimal variant (X-ray or CT measurements, porosity analysis, …etc.).
Author Response
Point 1: The paper presents a CAD/CAM/CAE analysis of high-pressure die casting process using a kitchen cooking grill as case study. Standard 3D techniques (UNIGRAPHICS and MAGIC) are applied for generating three variants of runners which are further analyzed by commercially available AnyCasting software regarding fluid flow, filling and solidification process. The comparison of three variants is correct and the advantages of third variant against others is clearly documented. The value of the paper is the good application of available tools for a given task, but without any scientific added value and experimental validation. I am suggesting for the authors to extend the paper with experimental results of optimal variant (X-ray or CT measurements, porosity analysis, …etc.).
Point 1 Response: Thank you for your comment. Yes, your comment is correct. It would be better to convince our results if I provided the actual casting parts, X-ray, or CT measurements. However, it is too much cost for us to fabricate a mold to validate the simulation results. To improve the paper's quality, there are some modifications added in introduction, sections 2.1 and 3.2, and the conclusion. From my previous experience of other projects, the simulation result itself seems to be enough information for a mold-maker to save cost and time while producing the casting mold.
Point 2: Moderate English changes required
Point 2 Response: The paper is reviewed carefully to improve the English quality. On your point, English language editing is still needed. I will request the English language editing from MDPI to improve the English qualification of the article after clearing other issues.

Round 2
Reviewer 1 Report
The revised MS contains only superficial modifications to the current text and do not add to the scientific value of the work. The results and analysis sections remains very weak and in a "show-and-tell" fashion. The reviewer's original decision for rejection still stands.
Author Response
Point 1: There scientific or engineering contents is lacking on the MS. The cursory introduction of the governing equations neglected additional, more complex processes such as phase change (solidification) and subsequent flow in the partially solidified porous structure.
Point 1 Response: Thank you for your comment. Yes, I do agree with your opinion. Solidification is a complex problem of phase transformation involving the absorption and release of heat, and also, the phenomenon has nonlinear boundary conditions between different phases. Some modifications are added in sections 2.1 and 3.2.
Point 2: Without comparison with observation, it is very difficult to judge the skill of the model.
Point 2 Response: Thank you for your comment. Yes, I do agree with your opinion. It does not seem easy to judge the simulation results without comparing the actual casting parts, X-ray, or CT measurements. However, the simulation result seems to be enough information for a mold-maker to save cost and time while producing the casting mold from my previous experience of other projects.
Point 3: This MS is of very low scientific or engineering merit, and is one the reviewer would expect to see in CAE software user conferences.
Point 3 Response: Thank you for your comment. Similar types of projects can be seen in the user conference, but the data in the paper are the results of my own work.
Point 4: The revised MS contains only superficial modifications to the current text and do not add to the scientific value of the work. The results and analysis sections remains very weak and in a "show-and-tell" fashion.
Point 4 Response: Thank you for your comment. I agree that scientific value is not high, but the results and analysis of the paper were not produced just in a "show-and-tell" way. I have derived the best results in a given environment.

Reviewer 3 Report
The revised version has been significantly improved by additional information related the literature survey and the theoretical background of fluid flow and heat transfer. The analysis of shrinkage process has been extended and conclusions were widened with new valuable remarks. Authors do not mention the possibility of experimental validation – it is obvious that this needs more time and it might be the topic of a new publication.
Author Response
Point 1: Authors do not mention the possibility of experimental validation – it is obvious that this needs more time and it might be the topic of a new publication.
Point 1 Response: Thank you for your comment. Yes, I do agree with your opinion. It does not seem easy to validate the simulation results without comparing the actual casting parts, X-ray, or CT measurements. However, the simulation result seems to be enough information for a mold-maker to save cost and time while producing the casting mold from my previous experience of other projects. And also I have derived the best results in a given environment.
By the way, some modifications are added in sections 2.1 and 3.2 to improve the paper quality.
Thank you very much and your consideration.
